# Multi-Omic Biomarkers for Patient Stratification in Sjogren’s Syndrome—A Review of the Literature

**DOI:** 10.3390/biomedicines10081773

**Published:** 2022-07-22

**Authors:** Lucia Martin-Gutierrez, Robert Wilson, Madhura Castelino, Elizabeth C. Jury, Coziana Ciurtin

**Affiliations:** 1Centre for Rheumatology Research, Division of Medicine, University College London, London WC1E 6JF, UK; l.martin-gutierrez@ucl.ac.uk (L.M.-G.); e.jury@ucl.ac.uk (E.C.J.); 2Department of Rheumatology, University College London Hospitals NHS Trust, London NW1 2PG, UK; robert.wilson24@nhs.net (R.W.); madhura.castelino@nhs.net (M.C.); 3Centre for Adolescent Rheumatology Versus Arthritis, Division of Medicine, University College London, London WC1E 6JF, UK

**Keywords:** Sjogren’s syndrome, patient stratification, clinical relevance, multi-omics

## Abstract

Sjögren’s syndrome (SS) is a heterogeneous autoimmune rheumatic disease (ARD) characterised by dryness due to the chronic lymphocytic infiltration of the exocrine glands. Patients can also present other extra glandular manifestations, such as arthritis, anaemia and fatigue or various types of organ involvement. Due to its heterogenicity, along with the lack of effective treatments, the diagnosis and management of this disease is challenging. The objective of this review is to summarize recent multi-omic publications aiming to identify biomarkers in tears, saliva and peripheral blood from SS patients that could be relevant for their better stratification aiming at improved treatment selection and hopefully better outcomes. We highlight the relevance of pro-inflammatory cytokines and interferon (IFN) as biomarkers identified in higher concentrations in serum, saliva and tears. Transcriptomic studies confirmed the upregulation of IFN and interleukin signalling in patients with SS, whereas immunophenotyping studies have shown dysregulation in the immune cell population frequencies, specifically CD4^+^and C8^+^T activated cells, and their correlations with clinical parameters, such as disease activity scores. Lastly, we discussed emerging findings derived from different omic technologies which can provide integrated knowledge about SS pathogenesis and facilitate personalised medicine approaches leading to better patient outcomes in the future.

## 1. Introduction

Sjögren’s syndrome (SS) is an autoimmune rheumatic disease (ARD) characterised by a chronic inflammatory process associated with lymphocytic infiltrate affecting the exocrine glands. The disease has significant heterogeneity in clinical presentation according to age at disease onset, type of organ involvement, as well as serological features and response to therapy [1,2]. When the disease occurs on its own, it is called primary SS (pSS), while when it accompanies other autoimmune conditions, it is defined as secondary SS (sSS). Various classification criteria have been used to define pSS and exclude mimicking pathology, with the most recent ones being the data and consensus-driven American College of Rheumatology/European League Against Rheumatism Classification Criteria proposed in 2016 [3].

There are currently no universally accepted classification criteria for sSS and some experts argue that making a distinction between pSS and sSS is not adequate anymore, as both phenotypes represent the same disease [4]. Moreover, the classification criteria validated in adults have minimal utility in SS with childhood-onset (defined as disease onset before the age of 18 years), as the disease presentation in children and young people, although rare, is different [5]. This, in addition to the lack of validated classification criteria for childhood-onset SS, further limits the research opportunities for younger people affected by this disease [6].

The disease manifestations vary among patients; some have predominant exocrine glandular involvement leading to dryness, which is the hallmark symptom of the disease. Glandular involvement manifests as dry mouth (xerostomia), dry eyes (xerophthalmia), dry skin (xerosis cutis), as well as vaginal dryness, dry cough, pancreatic dysfunction, salivary gland inflammation/enlargement (e.g., parotitis), etc., while patients with extra-glandular involvement can experience frequent musculoskeletal, haematological, and rarely hepatic, renal, pulmonary, cardiac, peripheral or central nervous system manifestations, as well as less specific symptoms of fatigue (common) or fever and lymphadenopathy (less common) [7]. Clinical presentation usually guides the disease management, which is largely symptomatic for glandular manifestations and involves the use of immunosuppressive treatment approaches in patients with more severe organ involvement [8]. The evidence for the efficacy of various therapies currently recommended for the management of SS is modest overall [9,10], emphasising the need for better research.

As a direct consequence of the disease pathogenesis being centred around the process of autoimmune epithelitis, powered by the interplay between the cells of the innate and adaptive immune systems, and activated by interferons and other pro-inflammatory cytokines leading to chronic immune activation in a host with genetic susceptibility [11], various disease fingerprints can be identified from the peripheral blood, as well as serum, saliva, tears and salivary gland biopsies.

Significant progress has been achieved recently in clinical research in terms of better patient clinical and molecular characterisation [12], but despite this, the management of this condition remains challenging because of patient heterogeneity and various limitations of the way the disease activity and response to treatment are measured. These aspects very likely contribute to the lack of significant treatment advances in SS, despite preliminary signals of the efficacy of various biologic agents in clinical trials [13,14,15]. Better research into disease pathogenesis and distinct clinical and molecular phenotypes will hopefully enable better patient selection for available therapies as well as new target discoveries.

## 2. Materials and Methods

This review aimed as identifying the main papers published since 2000 investigating multi-omics (cytokine profiling in serum, tears, saliva, immunophenotyping, genomic, transcriptomic and metabolomic) studies in SS in an effort to identify distinct patient groups (endotypes) which can inform meaningful stratification for better disease characterisation and improved treatment strategies. Publications selected for this review followed these inclusion criteria: Sample size higher than 10, the inclusion of age and gender-matched healthy controls, and data on at least one omic analysis in any biologic sample relevant to SS (blood, serum, tears, saliva, salivary gland biopsy) and published in English. We presented the most informative papers found in the literature in tables, summarising the study design, sample size, control groups, main findings and their clinical relevance.

## 3. Results

### 3.1. Multi-Omic Biomarkers for Patient Stratification

#### 3.1.1. Tear Biomarkers

One of the main symptoms of pSS is dry eye (xeropththalmia) as a result of lymphocytic infiltration of the lacrimal glands. Tears represent a valuable biological sample resource because of their proximity to the site of glandular inflammation and they might contain biomarkers that could help us to understand the pathogenesis of pSS, improve its diagnosis and have therapeutic implications.

Recently, several studies have concentrated their efforts on identifying those biomarkers through either cytokine, metabolomic or proteomic tear profiling (Table 1).

In this context, Chen et al. [16] determined the cytokine profile of tears, measured by a 27-plexcytokine assay in 29 pSS patients and 20 gender/age matched controls (non-SS sicca subjects and healthy controls—HCs). Elevated levels of pro-inflammatory cytokines, such as interleukin (IL)-1 receptor agonist (ra), IL-2, IL-17A, interferon (IFN)-γ, Macrophage inflammatory protein-1-β (MIP-1b), and Rantes (Regulated upon Activation, Normal T Cell Expressed and Presumably Secreted) and anti-inflammatory interleukin 4 IL-4, were found in pSS patients compared to controls. Interestingly, higher cytokine levels correlated positively with eye dryness severity and negatively with Schirmer’s test which measures the volume of tears secreted over 5 min [16]. These findings are validated by another study Willems et al., which also found an increased concentration of IFN-γ, tumor necrosis factor alfa (TNF-α), IL-2, IL-4, IL-6, IL-10, IL-12p70 and IL-5 in tears of pSS patients compared to HCs [17]. Moreover, they also verified the negative correlation between the Schirmer test and the concentration of IL-2, IL-4, IL-10 and IL-12p70 in tears.

Tears are composed of water, electrolytes, mucins and hundreds of different proteins and metabolites. Urbanski et al. identified a metabolic signature of tears comprising nine metabolites specific to pSS, compared to patients with non-immune dry eye disease [18]. Metabolomic quantification by mass spectrometry and liquid chromatography showed that three metabolites, serine, aspartate and dopamine, had lower concentrations whereas six lipids (including pro-inflammatory lysophosphatidylcholine [19], sphingomyelin, and phoshatidylcholine diacyl) had increased concentrations in pSS patients compared to the non-pSS Sicca controls. Moreover, age, sex, use of anticholinergic drugs, or the presence of anti-Ro/SSA antibodies did not influence the association between the metabolomic signature and the pSS status, suggesting that it is a true disease signature.

Tear proteomic analysis by Das et al. [20] using high performance liquid chromatography (HPLC) and mass spectrometry revealed the upregulation of 83 proteins and downregulation of 112 proteins in pSS patients compared to HCs. Enrichment pathway analysis of upregulated proteins included leukocyte trans-endothelial migration, protein-lipid complex remodelling and collagen catabolic pathway. On the other hand, the analysis of downregulated proteins indicated that pathways, such as glycolysis and amino acid metabolism, were diminished in tears from pSS patients. The relationship between proteomic biomarkers and clinical outcomes was not explored.

**Table 1 biomedicines-10-01773-t001:** Examples of studies investigating potential Sjögren’s Syndrome biomarkers in tears.

Reference	Type of Study/Samples/Methods	Number (N) of pSS Patients and HCs Age (Mean ± SD)	Disease Signature Identified	Correlations with Clinical Outcomes
BIOMARKERS IN TEARS				
**Cytokine profiling**				
Chen et al., 2019 [16]	Cross-sectional Tear strips for Schirmer I test Unstimulated (UWS) and stimulated (SWS) saliva samples Method: Cytokine 27-plex Assay	N = 29 pSS 56.8 ± 13.0 years N = 20 sicca (non-SS) controls 51.7 ± 10.6 years N = 17 HCs 45.4 ± 10.9 years	Increased IL-1ra, IL-2, IL-4, IL-17A, IFN-γ, MIP-1b, and Rantes in pSS vs. non-SS/HCs (*p* < 0.05).	Increased dry eye severity level and ocular surface staining correlated with increased tear cytokine levels, except for IP-10. Negative correlations between Schirmer’s test and tears IL-1ra, IL-2, IL-4, IL-8, IL-12p70, IL-17A, IFN-γ, MIP-1b, and Rantes (r = 0.26–0.61, *p* < 0.05).
Willems et al., 2021 [17]	Cross-sectional Tear samples Method: LUNARIS™ BioChip	N = 12 pSS 41.7 ± 13.3 years N = 13 HCs 43.0 ± 13.8 years	Tears: Increased I FN-γ, TNF-α, IL-2, IL-4, IL-6, IL-10 and IL-12p70 (left eye) and IL-5 (right eye) in pSS compared to non-SS and HCs (*p* < 0.005).	Schirmer test correlated to IL-2 (r = −0.702), IL-4 (r = −0.769), IL-10 (r = −0.839) and IL-12p70 (r = −0.753) left eye levels; IL-10 directly correlated with SPEED test score (r = 0.722; *p* = 0.0001) as well as NIKBUT score (r = −0.705; *p* = 0.00002).
**Metabolomic profiling**				
Urbanski et al., 2021 [18]	Cross-sectional Tear strips for Schirmer I test Method: mass spectrometry/liquid chromatography	N = 40 female pSS 63 years N = 40 non pSS sicca controls 58 years	9 metabolites (serine, aspartate; dopamine and six lipids) defined a tear pSS metabolomic signature (ROC-AUC = 0.83) PCA analysis showed that 2 PC explained 74.5% variance defined by 8/9 metabolites; the six lipids were distributed in the PC 1 and the amino acids in the second one.	The association between the metabolomic signature and the pSS status was not altered by age, sex, use of anticholinergic drugs or presence of anti-SSA antibodies
**Proteomic profiling**				
Das et al., 2021 [20]	Cross-sectional Tears, Tear washes Saliva Cryopreserved parotid gland biopsy samples Methods High performance liquid chromatography HPLC/mass spectrometry MS shotgun proteomics analysis Biopsy staining with anti-PRG4 mAb Bead-based immunoassay using the AlphaLISA	Tears N = 22 pSS (F:M = 10:1) 60.0 ± 16.5 years N = 20 HCs (F:M = 13:7) 31.2 ± 11.4 years Tear washes N = 14 pSS (F:M = 13:1) 59.5 ± 12.0 years N = 29 HCs (F:M = 17:12) 34.1 ± 14.2 years	Tears: 83 upregulated and 112 unique downregulated proteins in pSS vs. HCs. Enriched pathways in pSS: leukocyte trans-endothelial migration, protein-lipid complex remodelling and collagen catabolic. Enriched pathways in HCs: glycolysis/gluconeogenesis and glycolysis in senescence, amino acid metabolism and VEGFA/VEGFR2 signalling pathway. Overall, there was a loss of glycolysis and metabolism but an elevation of immune processes in pSS tears samples. PRG4 in tear washes was significantly decreased in pSS (*p* < 0.01).	Not explored

Legend: pSS—primary Sjögren’s Syndrome, HC—healthy controls, Rantes—Regulated upon Activation, Normal T Cell Expressed and Presumably Secreted, MIP-1b—Macrophage Inflammatory Proteins, IFN—interferon, IL—interleukin, VEGFA—Vascular endothelial growth factor A, VEGFR2—VEGF receptor 2, PRG4—Proteoglycan 4.

#### 3.1.2. Saliva Biomarkers

Dry mouth (xerostomia) is a key symptom of pSS occurring in more than 95% of patients as a consequence of autoimmune destruction of salivary glands [21,22]. Salivary gland pathology detected by salivary gland biopsy is included in the classification criteria for pSS, and in many patients, this is an essential diagnostic and prognostic tool. However, this is an invasive procedure that could lead to local complications in a minority of cases, [23] whereas the collection of saliva for research purposes is, in contrast, a non-invasive procedure. Biomarkers found in saliva could potentially reflect the pathogenesis of this disease. Therefore, many researchers have been aiming to identify those lately (Table 2).

One of the few studies examining the cytokine profile of unstimulated saliva of pSS patients using the Luminex platform found an increase in IFN-γ, IL-1, IL-4, IL-10, IL-12p40, IL-17, and TNF-α levels in pSS patients compared to non-SS and HCs [24]. Moreover, IL-6 levels were higher in pSS compared to HCs. Notably, unstimulated saliva flow rate correlated with INF-γ/IL-4 ratio and salivary gland biopsy focus score (the number of inflammatory infiltrates of at least 50 cells present in 4 mm^2^ of salivary gland area) correlated with TNF-α/IL-4 ratio in pSS, suggesting a predominant Th1 saliva signature. Years later, Chen et al. [16] reported enhanced levels of IP-10 (Interferon gamma-induced protein 10 or C-X-C motif chemokine ligand 10, CXCL10) and MIP-1α in saliva samples from pSS compared to HCs and a negative correlation between MIP-1α levels and both unstimulated whole saliva as well as the stimulated whole saliva flow rates.

Metabolomics analysis of saliva identified a total of 41 metabolites reduced in pSS patients compared to HCs [25]. Principal component analysis (PCA) revealed that saliva from pSS patients had less biological diversity compared to HCs. Two distinct groups of pSS patients were identified based on their metabolomic profile: the only clinical differences between the groups were older age and the presence of major salivary gland glanditis in one group compared to the other. Recently, a longitudinal study by Herrala et al. [26] investigated changes in the levels of salivary metabolites in pSS and HCs using proton nuclear magnetic resonance (NMR) spectroscopy over 20 weeks. Choline, taurine, alanine, and glycine were the most significantly different metabolites, all of them were found in higher concentrations in saliva samples from pSS patients than in HCs. Compared to the baseline of the HCs, choline was significantly elevated at each time point, taurine and glycine were significantly higher at weeks 1, 10 and 20, whereas alanine was higher at weeks 10 and 20, suggesting that the distinct saliva metabolic signature is relatively stable over time.

Saliva proteomic analysis by Delaleu et al. [27] using a multiplex capture antibody-based assay identified 61 differentially expressed proteins in pSS vs. non-SS controls including rheumatoid arthritis (RA) and HCs samples. Interestingly only one protein, fibroblast growth factor (FGF)-4, was found at a lower concentration in pSS while 60 different proteins were present at higher concentrations compared to controls. This comprehensive analysis recognised a proteomic signature based on the following proteins: clusterin, IL-5, FGF-4 and IL-4. The proteomic signature could correctly identify pSS patients with an accuracy of 93.8% and non-SS patients with an accuracy of 100%. However, none of the protein biomarkers correlated with saliva flow rates in pSS. A more recent study by Das et al. [20] identified the upregulation of 104 proteins and downregulation of 42 proteins in pSS compared to controls. Some enriched pathways in patients’ saliva included JAK-STAT signalling after IL-12 stimulation, superoxide metabolic process and phagocytosis.

#### 3.1.3. Potential pSS Biomarkers in Peripheral Blood

It is well known that pSS is characterised by an imbalance of immune cell types, including a loss of T cell tolerance and autoreactivity, increased infiltration of exocrine gland tissues, contributing to the inflammatory microenvironment, as well as B cell activation, which is crucial for ectopic lymphoid structure and germinal centre formation, which eventually leads to the irreversible glandular damage [29]. Table 3.

Mingueneau et al. [30] published a fascinating study whereby, using mass spectrometry and immunochemistry in paired blood and salivary gland biopsies, a SS disease signature was uncovered. These findings highlighted the presence of activated CD8^+^ T cells, terminally differentiated plasma cells, and activated epithelial cells in biopsies, whereas in blood samples they observed a cell signature of low numbers of CD4^+^ T cells, memory B cells, plasmacytoid dendritic cells and high numbers of activated CD4^+^, CD8^+^ T cells and plasmablasts. The blood signature observed correlated with clinical parameters and enabled patient stratification into different endotypes with distinct disease activity and degrees of glandular inflammation. In line with this result, Van der Kroef et al. [31] also observed reduced frequencies of memory B cells and plasmacytoid dendritic cells and increased frequencies of activated HLA-DR CD4^+^ and CD8^+^ T cells in pSS patients compared to HCs.

In 2021, Szabó et al. [32] published an article whose aim was to investigate whether the distribution of B cells in pSS could be affected by a change in the balance of circulating T follicular helper (Tfh) cell subsets and follicular regulatory T cells. Utilising multicolour flow cytometry, they discovered that pSS patients had a significant increase in activated Tfh cells compared to HCs. Interestingly, anti-La/SSB-positive patients had a higher frequency of T follicular regulatory cells compared to seronegative patients. In the B-cell compartment, they observed that memory B cells were decreased, and transitional and naïve B cells were significantly increased. Lastly, they identified a positive correlation between the proportion of activated Tfh cells and both the levels of anti-La/SSB autoantibody and the serum IgA titre. Moreover, they demonstrated the frequency of pro-inflammatory Tfh1 cells correlated positively with levels of serum IgG and anti-LA/SSB autoantibody, suggesting the potential implication of various immune cell subsets in the disease pathogenesis through correlations with serological markers.

Martin-Gutierrez et al. [33] identified an immune signature derived from the analysis of 29 different cell subsets including B and T cells, which was driven by five distinct cell subsets: transitional Bm2′ cells, late memory Bm5 cells, IgD-CD27-B cells, and CD8^+^ naive and CD8^+^ Tem which differentiated between pSS patients and matched HCs. Moreover, they identified a shared immunological profile across three disease phenotypes: systemic lupus erythematosus (SLE), pSS and SLE associated with SS. By applying machine learning approaches, they identified two patient endotypes based on immune cell alterations, irrespective of the underlying diagnosis, suggesting significant pathogenic commonalities between these three disease groups. Notably, correlations were found between clinical manifestations and the frequencies of the immune cell subsets driving the stratification. CD8^+^ and CD4^+^ T cell subsets and B cell populations correlated with the erythrocyte sedimentation rate (ESR) in pSS patients whereas haemoglobin levels correlated with the frequency of CD8^+^central memory T cells. Disease damage scores also correlated with the frequency of CD8^+^ TEMRA (effector memory T cells re-expresses CD45RA)cells, CD8^+^ responder cells (CD25-CD127^+^) and CD8^+^CD25-CD127-T cells.

Single-cell RNA sequencing of peripheral blood mononuclear cells (PBMCs) identified the expansion of CD4^+^ cytotoxic T lymphocytes and a population of CD4^+^ T cells highly expressing the T cell receptor Alpha Variable 13-2gene, in pSS patients compared to HCs [34]. Pathway enrichment analysis revealed upregulation of genes involved with type I and II interferon signalling, TNF family signalling and antigen processing and presentation in pSS patients. Using flow cytometry, it was confirmed the percentages of CD4^+^ Granzyme B^+^ T cells in the CD4^+^ T cell populations were significantly higher in pSS patients compared to the HCs. No correlations were found between the frequencies of CD4^+^ T cells and clinical or serological parameters, including the disease activity index ESSDAI (EULAR Sjögren’s syndrome (SS) disease activity index), ESR levels, or the presence of anti-Ro antibodies.

Disease-associated biomarkers can be detected in serum through proteomic or metabolomic technologies. Serum concentrations of proteins in pSS patients and HCs were measured by a high-throughput proteomic assay in a recent publication [35]. Using this complex assay 1110 proteins were quantified and, from those, 82 were found to be differentially expressed in pSS patients. Significant correlations between nine differently expressed serum proteins and the ESSDAI score were found. Using a second cohort of pSS patients, five proteins including CXCL13, TNF-receptor 2, CD48, B-cell activating factor (BAFF), and PD-L2 (Programmed cell death ligand 2) were validated as pSS-associated biomarkers. Another study investigated which serum protein biomarkers, measured by Bio-Plex, could distinguish pSS from other autoimmune diseases, such as SLE and RA [36]. Out of 63 proteins, they were able to identify eight and four proteins that could differentiate pSS from SLE and RA, respectively. A combination of four different proteins: BDNF (Brain Derived Neurotrophic Factor), I-TAC/CXCL11, soluble (s) CD163 and Fractalkine/CX3CL1 was identified as a pSS protein signature as it could discern pSS from other autoimmune diseases. A negative correlation between ESSDAI score and serum sCD163 concentrations was found.

Different reports [37,38,39] have also focused on analysing changes in serum metabolites by different techniques, such as mass spectrometry to find new molecules that could play a role in the pathogenesis of pSS and could become new drug targets [39]. Using a non-targeted gas chromatography-mass spectrometry (GC-MS) serum metabolic profile, the authors detected 21 metabolites that differentiated between pSS patients and controls, with 18 out of 21 metabolites further validated in another cohort. Two metabolites, stearic acid and linoleic acid had the adequate discriminatory capacity to separate pSS patients from HCs and correlated with clinical parameters, such as C-Reactive Protein (CRP), ESR, IgG, anti-Ro/SSA, anti-La/SSB, antinuclear antibodies, IgA and rheumatoid factor.

**Table 3 biomedicines-10-01773-t003:** Examples of studies investigating potential Sjögren’s Syndrome biomarkers in saliva.

Reference	Type of Study/Samples/Methods	Number (N) of pSS Patients and HCs Age (Mean ± SD)	Disease Signature Identified	Correlations with Clinical Outcomes
BIOMARKERS IN PERIPHERAL BLOOD				
**Immunophenotype profiling**				
Mingueneau et al., 2016 [30]	Cross-sectional PBMC samples, a subset of paired SG biopsies Method: CyTOF, immunohistochemistry	N = 49 pSS 53 years N = 45 sicca (non-SS) and HCs 54 years	SG biopsies: increased activated CD8^+^ T cells, terminally differentiated plasma cells, and activated epithelial cells Blood: 6-cell signature: decreased CD4, memory B-cells, plasmacytoid dendritic cell, and increased activated CD4 and CD8 T cells and plasmablasts.	The blood cellular components correlated with clinical parameters clustered patients into subsets with distinct disease activity and glandular inflammation.
Van der Kroef et al., 2020 [31]	Cross-sectional, PBMC samples Methods: Luminex, CyTOF	N = 88 SSc 54 years N = 31 SLE 43 years N = 23 pSS 56 years N = 44 HCs 50 years	pSS patients have increased HLA-DR CD4^+^ and CD8^+^ frequencies and reduced memory B cells and pDCs compared to HCs.	Not explored in pSS
Szabó et al., 2021 [32]	Cross-sectional PBMCs samples Methods: flow cytometry, functional analysis and ELISA.	N = 38 pSS 54 years N = 27 HCs 46 years	pSS patients showed a significant increase in activated T follicular helper cells. Frequencies of T follicular regulatory cells were increased in autoantibody La positive patients compared to seronegative pSS. Transitional and naïve B cells increased, memory B cells decreased,	The percentage of activated T follicular helper cells showed a positive correlation with the levels of anti-La/SSB autoantibody and with serum IgA titre. Frequency of Tfh1 positive correlation with levels of serum IgG and anti-LA/SSB autoantibody.
Martin-Gutierrez et al., 2021 [33]	Cross-sectional PBMC samples Method: Flow cytometry/ML	N = 45 pSS 59 (30–78) N = 29 SLE 48 (21–72) N = 14 SLE/SS 55 (26–56) N = 31 HCs	Patients with SS/SLE and SLE/SS shared immunological signatures. A signature comprising 5/29 immune cell subsets studied: transitional Bm2′ cells, late memory Bm5 cells, IgD-CD27-B cells, and CD8^+^ naïve and CD8^+^ Tem cells stratified patients	ESR correlated with 4 CD8^+^ T cell, 3 CD4^+^ T cell and 2 B cell subpopulations, which drove patient stratification. Hgb level correlated with % CD8^+^ Tcm cells. Disease damage scores across correlated with %CD8^+^ T cell, including CD8^+^CD25–CD127, CD8^+^ responder T cells, and CD8^+^ Temra cells
**Single-cell transcriptomic profile**				
Hong et al., 2021 [34]	Cross-sectional PBMCs Methods: scRNAseq and Flow cytometry	N = 10 pSS patients 48.8 years N= 10 HCs 33 years	Two subpopulations expanded in pSS: one expressing cytotoxicity genes (CD4^+^ CTLs cytotoxic T lymphocyte), and another highly expressing T cell receptor (TCR) variable gene (CD4^+^ TRAV13-2^+^ T cell). Total T cells significantly higher in pSS vs. HCs (*p* = 0.008). The IL-1β expression in macrophages, TCL1A in B cells, and IFN response genes in most cell subsets were upregulated in pSS. Susceptibility genes including HLA-DRB5, CTLA4, and AQP3 were highly expressed in pSS.	Correlation between the percentage of CD4^+^ CTLs and clinical characteristics, such as ESR), anti-SSA positive, and ESSDAI but no significant correlation was found.
**Serum proteomics**				
Nishikawa et al., 2016 [35]	Cross-sectional Methods: high-throughput proteomic analysis, ELISA.	Discovery cohort: N = 30 pSS 61 years N = 30 HCs 40 years Validation cohort: N = 50 pSS 60 years	A total of 82 (57 upregulated and 25 downregulated) serum proteins were differentially expressed in patients pSS vs. HCs. Enriched pathways: “extracellular region”, “chemokine signalling pathway”, “downstream of TNF-α”, “platelet activation”, and “platelet degranulation”. Nine proteins correlated with disease activity in the discovery cohort. In the validation cohort five proteins: CXCL13, TNF-R2, CD48, BAFF, and PD-L2 showed a correlation with ESSDAI, and therefore, were proposed as disease activity-associated biomarkers.	Serum concentrations of CXCL13, TNF-R2, and CD48 were positively correlated with that of immunoglobulin (Ig) G. TNF-R2 was negatively correlated with unstimulated salivary. BAFF was negatively correlated with the excretion rate in the submandibular gland.
Padern et al., 2021 [36]	Cross-sectional Methods: BioPlex, ELISA	N = 42 pSS 62.5 years N = 28 RA 60.5 years N = 25 SLE 40 years	Eight biomarkers could statistically discriminate samples from pSS versus SLE patients. Four could statistically discriminate pSS patients from RA patients. None of the studied biomarkers could simultaneously discriminate pSS from RA and SLE. We, therefore, determined the positive predictive value (PPV), sensitivity, and specificity of different combinations of BDNF, I-TAC/CXCL11, sCD163 and Fractalkine/CX3CL1 concentrations. These biomarkers were chosen because they were those most strongly associated with distinguishing pSS from the other AIDs.	Negative correlation between pSS activity according to the ESSDAI score and serum sCD163 concentrations.
**Serum metabolomic profiling**				
Xu et al., 2021 [39]	Cross-sectional Serum samples Method: non-targeted GC-MS	Discovery: 90 pSS patients, M:F = 1/10) 53 years 153 HCs (male/female: 1/10) 50.4 years Validation: 119 pSS, M:F = 1/10 52.9 years 143 HCs (M:F = 1/10 50.23 years	Increased alanine, tryptophan, glycolic acid, pelargonic acid, cis-1-2-dihydro-1-2-naphthalenediol, etc., and decrease in catechol, anabasine, 3-6-anhydro-D-galactose, beta-gentiobiose and ethanolamine in pSS patients vs. HCs. Stearic acid and linoleic acid distinguished pSS from HCs (ROC−AUC = 0.97−0.98)	Inflammatory markers, autoantibodies and Ig G levels correlated with various metabolite levels.

Legend: pSS—primary Sjögren’s Syndrome, HC—healthy controls, SLE—systemic lupus erythematosus, pDCs—plasmacytoid dendritic cell, Tfh—T follicular helper cells, ESR—Erythrocyte Sedimentation Rate, Hgb—Haemoglobin, AQP- Aquaporin, CTLA—Cytotoxic T-Lymphocyte Associated, CXCL-C-X-C motif chemokine ligand, TNFR—Tumour Necrosis factor receptor, BAFF—B-cell activating factor, PDL2—Programmed cell death 1 ligand 2, BDNF—Brain-derived neurotrophic factor, sCD163—soluble CD163, ESSDAI—EULAR Sjögren’s syndrome (SS) disease activity index.

#### 3.1.4. Genetic and Epigenetic Studies

Although the aetiology of SS is unknown, it is considered that different factors, such as environmental, genetic and epigenetic, contribute to the disease pathogenesis. In this context, several studies [40,41,42,43], have focused on finding genetic and epigenetic factors that could be associated with SS. Transcriptomics, genome-wide association studies (GWAS) to identify genomic variants that are statistically associated with a risk of suffering the disease and epigenetic studies to determine whether gene expression is active or inactive based on DNA methylation are widely used nowadays.

##### Multi-omic pSS Signatures

It is well known that the diagnosis and treatment of SS, is challenging due to the existing molecular and clinical heterogenicity, which reflects different disease stages, variable types of organ involvement, disease severity and treatment, as well as patient-specific factors, such as age, environmental exposures and comorbidities. Thus, recent research (Table 4) has been focused on integrating genomic/epigenomic, transcriptomic, proteomic, metabolomic and immunophenotype characterisation and clinical data to gain more knowledge about the disease pathogenesis as well as being able to classify patients into groups defined by their molecular pattern.

Integrated transcriptomic and serum proteomic data with an immune signature comprising 24 different cell populations highlighted the presence of a pSS gene signature driven by interferon genes as well as ADAMs (a disintegrin and metalloprotease) substrates [44]. Interestingly, the genomic regions coding the genes identified as part of the disease signature were predominantly hypomethylated, therefore, transcriptionally activated. In addition, the proteomic analysis revealed some correlations between ADAMs substrates and ESSDAI scores. Relevantly, the authors confirmed that CD8+ T cells, especially TEMRA, produced the signature observed. Similarly, transcriptomic and cytokine profiling of pSS patients allowed the stratification of pSS patients into three distinct clusters, defined by IFN-responsive and inflammation-associated genes [45]. Interestingly, patients belonging to the cluster with the strongest IFN and inflammation gene signature also had high ESSDAI scores and elevated levels of anti-Ro/SSA and La/SSB autoantibodies. This cluster was also defined by a high serum concentration of cytokines, such as LIGHT and Blys and chemokine CXCL13 [45]. Soret et al. [46] and Barturen et al. [47] independently validated some of these findings, by showing a pSS patient transcriptomic stratification also driven by IFN-related pathways. Interestingly, in both studies, a cluster of patients with low disease activity was identified, characterised by a transcriptomic profile similar to that of HCs. Soret et al. [46] did not detect any differentially expressed genes, single nucleotide polymorphisms (SNPs) or differences in B and T cells, monocytes, basophils, eosinophils and neutrophils frequencies in pSS patients with low disease activity when compared to HCs.

## 4. Discussion

The significant progress made by high-throughput technologies, increased effort for large-scale academic and industry research collaborations to facilitate external validation, and advancement of computer algorithms for big data integration and cluster analysis provide unprecedented opportunities for better patient classification, improved pathogenic characterisation, prediction and therapeutic opportunities across all autoimmune diseases. It is increasingly recognised the need for better quality research, including both pre-clinical and clinical validation to enable meeting the ultimate goal of achieving clinical utility and patient benefit. Despite the impressive therapeutic advances leading to licensing of many new targeted therapies in autoimmune rheumatic disease (ARDs), such as inflammatory arthritis or SLE [48], patients with SS do not benefit from the same range of therapeutic options available for other conditions, despite shared pathogenesis [49]. Many of the signals of efficacy from early phase clinical trials of various biologics investigated in SS have not been replicated in larger studies and research is ongoing [50,51].

Efforts have been made in improving the way the response to treatment in clinical trials of patients with SS is assessed [52], while current treatment recommendations expanded to targeted biologic treatment options despite of lack of large phase 3 clinical trials [10]. In addition, novel approaches, such as advocating for a molecular classification of SS to drive precision medicine strategies have been proposed [46], which suggests that the future of clinical research in SS will likely involve multi-omic characterisation of patients (Figure 1). In this respect, good quality, reproducible research involving large cohort collaborations to capture the disease heterogeneity, as well as facilitate the validation of disease signatures, is required to improve knowledge about SS pathogenesis and facilitate the much-needed therapeutic advances.

It is widely recognised that SS is associated with a genetic predisposition, similar to other ARDs, which has been confirmed in large GWAS studies which validated the associations with HLA, IRF5, STAT4 and BLK genetic loci, while also detecting novel susceptible loci [53]. The best characterised are the HLA genes, associated with an increased disease risk ranging from 1.85 to 3.41 as per a large meta-analysis [54]. Various non-HLA genes associated with the disease have also been described but very few have been validated across studies [11].

Research into the role of environmental factors and epigenetics currently supports the old hypothesis that a ubiquitous virus is a potential trigger for the mechanism of autoimmunity, with most data potentially implicating Epstein–Barr Virus (EBV), Human T cell Leukemia Virus-1 (HTLV-1), or Coxsackie virus in the development of SS [55], despite the lack of conclusive evidence for their causal role. The most well-defined epigenetic mechanisms likely to play a role in the pathogenesis of SS have been described as DNA methylation, histone modifications and non-coding RNAs [42].

SS is characterised by a pro-inflammatory environment and cytokine profiling of serum, tears and saliva identified a predominance of pro-inflammatory cytokines, such as MIP, IL-1, IFN-γ, TNF-α, IL-6, IL-12 or IL17 in various proportions, as well as increased anti-inflammatory molecules, such as IL-4 or IL-10. Some of these data have been validated across studies, while some of the biomarkers, including cytokine ratios suggesting a Th1 signature also correlated with clinically meaningful parameters, such as tear and saliva secretion [16,17,24].

Dysregulation of various immune cell populations has been hypothesised as one of the key factors implicated in disease pathogenesis. Immune profiling of patients with SS found distinct immune signatures in the salivary gland tissue compared to peripheral blood, as expected. The main players seem to be activated CD8^+^ T cells, terminally differentiated plasma cells, and activated epithelial cells in biopsies, whereas the peripheral blood signatures comprised high numbers of activated CD4^+^, CD8^+^ T cells. Although these signatures were not perfectly validated across various studies [30,33], some correlated with serological and clinical parameters, suggesting a potential clinical utility.

Proteomic analysis revealed distinct signatures in tears and saliva compared to the serum of patients SS, with the majority of signatures being able to differentiate, with high accuracy, SS patients from controls, and a few correlating with clinical meaningful parameters. Enriched pathway analysis also overlapped with some cytokine signature findings, such as the upregulation of the JAK-STAT signalling after IL-12 stimulation in saliva [27]. The protein patterns identified in saliva were associated with B cell immune responses, macrophage differentiation and T cell chemotaxis, which showed similarities with salivary gland histopathological features [27], suggesting a potential role for saliva analysis as a proxy measure of glandular inflammation.

Transcriptomic profiling of salivary gland tissue was characterised by the upregulation of IFN-α and IL-12/IL-18 signalling, as well as CD3/CD28 T cell activation, CD40 signalling in B-cells, as well as significant correlation with the IFN-α score in PBMCs [28], which shows similarities with proteomic profiles of saliva. IFN response genes were also upregulated in most cell subsets when assessed by single-cell blood transcriptomic analysis, highlighting the role of the IFN activation pathway in the pathogenesis of the disease. In terms of potential clinical implications, the IFNγ/IFNα mRNA ratio in salivary gland tissue was shown to have the best discriminative capacity for lymphoma development in patients with pSS [56]. Patient stratification based on transcriptomic signatures identified distinct clusters driven by IFN and B cell activation, as well as SNPs in HLA genes and epigenetic modifications including gene hypomethylation [46], all processes recognised as involved in the disease pathogenesis, although the clinical significance of patient stratification was less clear.

Metabolomic characterisation of serum, tears and saliva of patients with pSS identified distinct signatures with almost no overlap between various biologic fluids [18,26,39]. Further research exploring the inter-individual variability and its stability over time is required [26].

The power of integrating several omic technologies in the investigation of the disease fingerprints harnessed evidence for the role of cytotoxic CD8 T cells in the disease pathogenesis [44] as well as enabled the identification of inflammatory, lymphoid and IFN-driven patient clusters generated by a combination of the transcriptome, methylome and cytokine profilin [45,47]. Patient clusters driven by high IFN and pro-inflammatory signatures were also associated with high disease activity suggesting that these pathways are relevant to the disease pathogenesis.

## 5. Conclusions

Omic investigation of SS provides a valuable insight into the disease pathogenesis and patient molecular heterogeneity which has implications for SS prognosis and better management strategies to address the unmet patient needs. Further research into standardising technologies and validating findings across large patient populations, as well as further exploration of potential correlations with clinical significance, are required to establish which are the strongest molecular signals that could be potentially translated into research with patient benefit. Ultimately, integrating data provided by multiple omics analysis can provide the much-required complementary knowledge related to the interplay between genes, environment, immune cell activation and pro-inflammatory milieu which all sustain the pathogenic processes associated with SS.

Understanding how the disease’s natural course or treatment impacts these molecular signatures, as well as which pathways can be targeted by available and novel treatments will open a new era for research in SS.

## Figures and Tables

**Figure 1 biomedicines-10-01773-f001:**
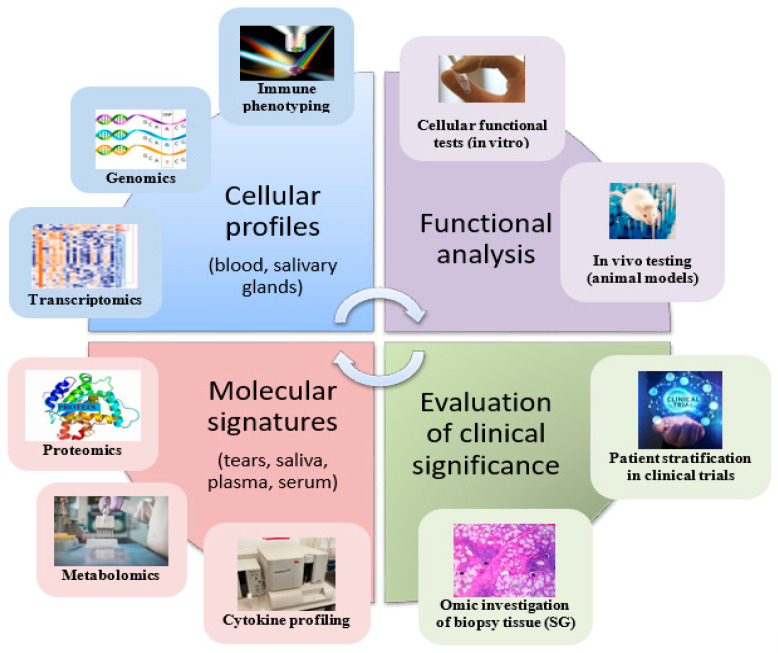
Potential multi-omic approaches taken in clinical research.

**Table 2 biomedicines-10-01773-t002:** Examples of studies investigating potential Sjögren’s Syndrome biomarkers in saliva/salivary glands.

Reference	Type of Study/Samples/Methods	Number (N) of pSS Patients and HCs Age (Mean ± SD)	Disease Signature Identified	Correlations with Clinical Outcomes
BIOMARKERS IN SALIVA/SALIVARY GLANDS				
**Cytokine profiling**				
Kang et al., 2011 [24]	Cross-sectional Unstimulated saliva samples Method: Luminex^®^ bead-based assay	N = 30 pSS 49.9 ± 9.0 years N = 30 sicca (non-SS) 51.5 ± 10.0 years N = 25 HCs 49.4 ± 9.5 years	Saliva: Increased IFN-γ, IL-1, IL-4, IL-10, IL-12p40, IL-17, and TNF-α levels in pSS vs. non-SS and HCs (*p* < 0.005). IL-6 levels higher in pSS vs. HCs (*p* = 0.011). INF-γ/IL-4; TNF-α/IL-4 higher in pSS vs. HCs (*p* = 0.028, *p* = 0.038, respectively).	No correlations were found between any salivary cytokine levels and clinical parameters. Unstimulated saliva flow rate correlated with INF-γ/IL-4 ratio (r = 0.411 *p* = 0.024) and focus score correlated with TNF-α/IL-4 ratio (r = 0.581, *p* = 0.023) in pSS, suggesting a predominant Th1 saliva signature.
Chen et al., 2019 [16]	Cross-sectional Tear strips for Schirmer I test Unstimulated (UWS) and stimulated (SWS) saliva samples Method: Cytokine 27-plex Assay	N = 29 pSS 56.8 ± 13.0 years N = 20 sicca (non-SS) controls 51.7 ± 10.6 years N = 17 HCs 45.4 ± 10.9 years	Saliva: increased IP-10 in pSS vs. non-SS/HCs. Both pSS and non-SS subjects had higher MIP-1α levels than HCs (*p* < 0.05).	UWS and SWS correlated negatively with MIP-1a saliva level (r = −0.276, *p* = 0.046 and r = −0.282, *p* = 0.040, respectively).
**Metabolomic profiling**				
Kageyama et al., 2015 [25]	Cross-sectional Unstimulated saliva samples Method: Gas chromatography mass spectrometry (GC-MS) analysis	N = 14 female pSS 44.2 ± 13.01 years N = 21 HCs 46.7± 10.24 years	41 of the metabolites were reduced in pSS patients compared to HCs (*p* < 0.05). Decreased glycine, tyrosine, uric acid and fucose in pSS vs. HCs in PCA analysis.	Patient stratification based on saliva metabolome depicted two groups: one younger (*p* = 0.082) and with a lower SG biopsy score (*p* = 0.014). Glycine levels differentiate between the two groups.
Herrala et al., 2020 [26]	Longitudinal study Stimulated saliva samples Method: proton nuclear magnetic resonance (1 H-NMR) spectroscopy	56 samples from N = 14 female pSS patients during four laboratory visits within 20 weeks. 48.6 years N = 15 HCs (mean age 49.8 years)	Increased choline in pSS patients at each time point (*p* ≤ 0.015), taurine at the last three time points (*p* ≤ 0.013), alanine at the last two time points (*p* ≤ 0.007) and glycine at the last time point (*p* = 0.005). Inter-individual variation observed for glycine (*p* ≤ 0.007, all time points), choline (*p* ≤ 0.033, three last time points) and alanine (*p* = 0.028, baseline).	Not explored
**Proteomic profiling**				
Delaleu et al., 2015 [27]	Cross-sectional Unstimulated whole saliva Methods: 187-plex capture antibody-based assay	Saliva N = 48 pSS (females) 47 years N = 24 non-SS cohort (12 RA patients + 12 HCs) 51 years	Significant differences in 61 biomarkers in pSS vs. controls (*p* < 0.001). FGF-4 levels lower in pSS (the only decreased protein). A biomarker signature comprising clusterin, IL-5, FGF-4, and IL-4 yielded accurate group prediction for 93.8% of pSS and 100% of non-SS controls classification.	No biomarkers correlated with salivary flow rates
Das et al., 2021 [20]	Cross-sectional Tears, Tear washes Saliva Cryopreserved parotid gland biopsy samples Methods High performance liquid chromatography HPLC/mass spectrometry MS shotgun proteomics analysis Biopsy staining with anti-PRG4 mAb Bead-based immunoassay using the AlphaLISA	Saliva N = 30 pSS (F:M = 22:8) 45.2 ± 14.6 years N = 10 HCs (F:M =5:5) 46.8 ± 14.5 years	Saliva: 104 upregulated and 42 downregulated proteins in pSS vs. HCs. Enriched pathways in pSS: JAK-STAT signalling after IL-12 stimulation, superoxide metabolic process and phagocytosis. Enriched pathways in HCs: neutrophil degranulation, negative regulation of peptidase activity; 2.3-fold increase in PRG4 in pSS (*p* < 0.05). PRG4 expression was found in both the serous acini and the striated duct on parotid gland biopsies (N = 4).	Not explored
**Salivary gland tissue transcriptomic profiling**				
Vertstappen et al., 2021 [28]	Cross-sectional Parotid and labial gland biopsy Methods: RNAseq—HiSeq 2500 System (Illumina). Multiplexed bead-based immunoassays for cytokine profiling Assessment of CD45-positive infiltrates on SG biopsies	N = 34 pSS with 51 paired (parotid and labial) biopsies 21 biopsy positive 13 biopsy negative 52 years N = 20 non SS sicca controls 17 biopsy negative 50 years	Parotid glands: 1041 up-regulated and 194 down-regulated DEG and labial glands: 581 and 43, respectively, between biopsy positive pSS and controls. The top 20 up-regulated genes in both tissues were mostly B-cell or T cell related. No significant differences between biopsy negative pSS and controls. Transcript expression levels correlated between parotid and labial glands (R^2^ = 0.86, *p* < 0.0001). Signatures enriched in biopsy-positive pSS compared with either biopsy-negative pSS or controls: IFN-α signalling, IL-12/IL-18 signalling, CD3/CD28 T cell activation, CD40 signalling in B-cells, double negative type-2 B-cells, and FcRL4^+^ B-cells. Strong correlation between the IFN-α score in PBMCs and SGs.	No difference in ESSDAI, unstimulated salivary flow or ESSPRI in patient DEG clusters.

Legend: pSS—primary Sjögren’s Syndrome, HC—healthy controls, IFN—interferon, IL—interleukin, TNF—Tumour necrosis factor, MIP—Macrophage Inflammatory Protein, FGF—Fibroblast growth factor, PRG—proteoglycan, FcRL-Fc Receptor Like, SGs—salivary gland, ESSDAI—EULAR Sjögren’s syndrome (SS) disease activity index, ESSPRI—EULAR Sjogren’s Syndrome Patient Reported Index.

**Table 4 biomedicines-10-01773-t004:** Examples of studies investigating potential Sjögren’s Syndrome biomarkers using multi-omic approaches.

Reference	Type of Study/Samples/Methods	Number (N) of pSS Patients and HCs Age (Mean ± SD)	Disease Signature Identified	Correlations with Clinical Outcomes
MULTIOMIC SIGNATURES				
Tasaki et al., 2017 [44]	Cross-sectional Methods: whole blood Transcriptomes microarrays, serum proteomes and peripheral immunophenotyping	N = 36 pSS patients 61 years N = 36 HCs 39 years	pSS gene signature predominantly involves the interferon signature including HERC5, EPSTI1 and CMPK2and ADAMs substrates. SGS was significantly overlapped with SS-causing genes indicated by a genome-wide association study as the regions that code genes in the SS gene signature were hypomethylated. Combining the molecular signatures with immunophenotypic profiles revealed that cytotoxic CD8 T cells were associated with SGS.	SGS positively correlated with the levels of autoantibodies, including anti-Ro/SSA and anti-La/SSB antigen–antibodies and serum IgG levels. Most ADAM substrates showed significant positive correlations with ESSDAI.
James et al., 2019 [45]	Cross-sectional Methods: RNAseq, Bioplex, ELISA, Luminex	N = 47 pSS patients 52 years	Three clusters of patients were identified based on transcriptomic analysis. No demographic differences between clusters. C1 weakest IFN signature and minimal activity of inflammation gene modules. C2 strongest IFN signature, strongest inflammation module. Higher ESSDAI scores. More patients presented anti-Ro and anti-La antibodies and higher levels. Higher levels of cytokines, such as LIGHT and Blys. CXCL13. C3 intermediate IFN signature, low activity of the inflammation modules.c3 higher levels of IL1, IL2 IL2RA.	C2 cluster presented higher ESSDAI scores
Soret et al., 2021 [46]	Cross-sectional Whole blood Methods: RNAseq, GWAS, Methylation and flow cytometry Serum sample Methods: Luminex, automated chemiluminescent immunoanalyzer (IDS-iSYS)	N = 304 pSS patients 58 years N = 330 HCs 53 years	Clustering of pSS samples based on transcriptomic data identified 4 different clusters (C1, C2, C3 and C4). C1 was enriched with IFN-related pathways, present an enriched up-regulated IFN signalling pathway; 35 SNPS were detected in genes associated with the immune system (HLA-DQB1, HLADQA1, HLA-DRA, HLA-C, HLA-G), signal transduction (NOTCH4), developmental biology (POU5F1), gene expression (DDX39B). Methylation in 87 genes. T cell lymphopenia. Increased in The IFNγ-induced protein (CXCL10/IP-10) as well as CCL8/MCP-2 and TNFα. IL-1 RII, was downregulated. No DEGs were noticed when comparing C2 to HCs. No SNPS were found in C2. Methylation of IFN genes MX1 and NLRC5. Frequency and the absolute number of T and B cells, monocytes, NK-like, DC, basophils, eosinophils, and neutrophils are similar to HCs. C3 was enriched with pathways related to B cell activation, and IFN signalling. SNPs detected in HLA-DQA, HLA-DRA (2 SNPs), BTNL2 and HCG23. Methylation in 56genes. Increased frequency of monocytes and lymphocytes. Increased in The IFNγ-induced protein (CXCL10/IP-10) as well as CCL8/MCP-2 and TNFα. IL-1 RII, was downregulated. C4 endotype with higher DEG including T and B activation, cytokine signalling and IL-15 production. The only SNPs identified in the intron LINC02571 gene and were previously associated with a risk for developing SLE. Methylation in 3000 genes. Decreased in B and T cells and monocytes. High percentage of neutrophils.	No statistically significant differences between the four clusters in ESSDAI or PGA mean scores. Statistically significant differences in the distribution of reported arthritis, rate of cancer history, coronary artery disease and chronic obstructive pulmonary disease were observed between the four clusters. Patients from C4 reported more severe clinical symptoms compared to the three other clusters. Some serological characteristics were significantly different across the four clusters, C1 and C3 have higher hypergammaglobulinemia, extractable nuclear antigen (ENA) antibodies, the presence of serum anti-SSA52/anti-SSA60 autoantibodies and higher circulating kappa and lambda free light chains (cFLC).
Barturen et al., 2021 [47]	Cross-sectional Follow-up Methods: Whole blood transcriptome and methylome	N = 955 cross-sectional patients with 7 autoimmune diseases 53.4 years N = 113 follow up patients. 47 years N = 267 HCs 46 years	Four clusters were identified and validated; 3 clusters represented inflammatory, lymphoid and interferon patterns; 1 cluster with low disease activity with no specific molecular pattern.	SLEDAI and ESSDAI scores were higher in all 3 clusters compared to the undefined cluster.

Legend: pSS—primary Sjögren’s Syndrome, HC- healthy controls, HERC5—HECT And RLD Domain Containing E3 Ubiquitin Protein Ligase 5, EPSTI1—Epithelial Stromal Interaction 1, CMPK2—Cytidine/Uridine Monophosphate Kinase 2, ADAM—A Disintegrin And Metalloprotease, SGS-Sjögren´s gene signature, IFN—interferon, CXCL-C-X-C motif ligand, SNPs—Single nucleotide polymorphisms, Notch—Neurogenic locus notch homolog protein, MX1—myxovirus resistance protein 1,NLRC5—NLR Family CARD Domain Containing 5, CCL8/MCP2—monocyte chemotactic protein-2, SLEDAI—Systemic Lupus Erythematosus Disease Activity Index, ESSDAI—EULAR Sjögren’s syndrome (SS) disease activity index.

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
