# Peer review of "Multi-Omic Biomarkers for Patient Stratification in Sjogren’s Syndrome—A Review of the Literature"

_biomedicines, 2022, doi:10.3390/biomedicines10081773_

Round 1

Reviewer 1 Report

SjÓ§gren’s syndrome (SS) is an autoimmune rheumatic disease characterized by a chronic inflammatory process associated with lymphocytic infiltrate affecting the exocrine glands. Understanding how the disease natural course and/or treatment impact their molecular signatures, as well as which pathways can be targeted by available and novel treatments has the potential of opening a new era for research in SS. The current manuscript by Martin-Gutierrez et al summarized recent multi-omic publications in order to identify biomarkers in tears, saliva and peripheral blood from SS patients that could be relevant for their better stratification aiming at improved treatment selection and hopefully better outcomes. They highlighted the relevance of pro-inflammatory cytokines and interferon as biomarkers identified in higher concentrations in serum, saliva and tears. They discussed emerging findings derived from different omic technologies which can provide integrated knowledge about SS pathogenesis and facilitate personalized medicine approaches leading to better patient outcomes in the future.

Overall, it’s an interesting topic, with comprehensive analysis of current status quo and mechanistic insights. The manuscript was well written.

Author Response

Thank you for you comments.

We proof-read the paper again and corrected all the typos related to the formatting process.  

I attached the version with tracked changes

Reviewer 2 Report

The manuscript reports an extensive review of papers published since 2000 on different biomarkers useful to stratify SS patients.

The paper is clearly written and can be informative for the readers.

The paper is not a metanalysis, but it would be more informative to detail how the mentioned papers were selected. 

I would also like to suggest the quote this paper (doi: 10.1038/s41598-021-01324-0). 

Author Response

Thank you for your suggestion. We added details about the selection of papers and the reference suggested. 

I attached the manuscript with tracked changes in response to Reviewer 1 and the clean version in response to Reviewer 2 as no possibility to attach two files. 
